

# SITool (v1.0) - a new evaluation tool for large-scale sea ice simulations: application to CMIP6 OMIP

Xia Lin[1,2], François Massonnet[1], Thierry Fichefet[1], Martin Vancoppenolle[3]

[1]Georges Lemaître Centre for Earth and Climate Research, Earth and Life Institute, Université
catholique de Louvain, Louvain-la-Neuve, 1348, Belgium

[2]Southern Marine Science and Engineering Guangdong Laboratory (Zhuhai), Zhuhai, 519000, China

[3]Laboratoire d'Océanographie et du Climat, CNRS/IRD/MNHN, Sorbonne Université, 75252, Paris,
France

*Correspondence to*: Xia Lin (xia.lin@uclouvain.be)

**Abstract.** The Sea Ice Evaluation Tool (SITool) described in this paper is a performance metrics and
diagnostics tool developed to evaluate the skill of bi-polar model reconstructions of sea ice
concentration, extent, edge location, drift, thickness, and snow depth. It is a Python-based software and
consists of well-documented functions used to derive various sea ice metrics and diagnostics. Here, the
SITool version 1.0 (v1.0) is introduced and documented, and is then used to evaluate the performance of

global sea ice reconstructions from nine models that provided sea ice output under the experimental
protocols of the Coupled Model Intercomparison Project 6 (CMIP6) Ocean Model Intercomparison
Project with two different atmospheric forcing datasets: the Coordinated Ocean-ice Reference
Experiments version 2 (CORE-II) and the updated Japanese 55-year atmospheric reanalysis (JRA55-
do). Two sets of observational references for sea ice concentration, thickness, snow depth, and ice drift

are systematically used to reflect the impact of observational uncertainty on model performance. Based
on available model outputs and observational references, the ice concentration, extent, and edge
location during 1980-2007, as well as the ice thickness, snow depth, and ice drift during 2003-2007 are
evaluated. It is found that (1) in general, model biases are larger than observational uncertainties and
model performances are primarily consistent compared to different observational references, (2) By

changing the atmospheric forcing from CORE-II to JRA55-do reanalysis data, the overall performance
(mean state, interannual variability and trend) of the simulated sea ice areal properties in both
hemispheres, as well as the mean ice thickness simulation in the Antarctic, the mean snow depth and ice
drift simulations in both hemispheres are improved, (3) the simulated sea ice areal properties are also
improved in the model with increased spatial resolution, (4) for the cross-metric analysis, there is no

link between the performance in one variable and the performance in another. The SITool is an open-
access version-controlled software that can run on a wide range of CMIP6 compliant sea ice outputs.
The current version of SITool (v1.0) is primarily developed to evaluate atmosphere-forced simulations
and it could be eventually extended to fully coupled models.



## 1 Introduction

Most regional and global climate models now include an interactive sea ice model, reflecting the reality that sea ice plays a fundamental role in the polar environment, by influencing air-sea-ice exchanges, atmospheric and oceanic processes, and climate change. Large inter-model spread exists in the performance of sea ice simulations in the Coupled Model Intercomparison Project 5 (CMIP5) for both the Arctic and Antarctic (Massonnet et al., 2012; Stroeve et al., 2012, 2014; Turner et al., 2013; Zunz et

al., 2013; Shu et al., 2015). Some improvements are identified in the CMIP6 models: (1) a more realistic estimate of sea ice loss for a given amount of $CO_2$ emissions and global warming in the Arctic (Notz et al., 2020), (2) reduced inter-model spreads in summer and winter ice area and improved ice concentration distribution in the Antarctic (Roach et al., 2020), (3) lower inter-model spreads in the mean state and trend of both the Arctic and Antarctic ice extents (Shu et al., 2020). However, sea ice

projections and evaluations are still not systematic and to date, no tool allows precise tracking of sea ice model performance through time from one version to the next. The Earth System Model Evaluation Tool (ESMValTool) has been developed for routine evaluation of climate model simulations in CMIP including many components of the Earth system (Eyring et al., 2016, 2020). While it is an efficient tool to obtain a broad view on the overall performance of a climate model, it is rather limited in terms of sea

ice diagnostics.

In this paper, we introduce the Sea Ice Evaluation Tool (SITool) for assessing large-scale sea ice simulations. SITool has been designed to describe inter-model differences both qualitatively and quantitatively and to help teams managing various versions of a sea ice model, detecting bugs in newly developed versions, or tracking the time-evolution of model performance. The SITool quantifies the

performance of sea ice model simulations by providing systematic and meaningful sea ice metrics and diagnostics on each sea ice variable with thorough comparisons to a set of observational references. Bi-polar performance metrics and diagnostics on ice coverage, drift, thickness and snow depth are provided from seasonal to multi-decadal time scales whenever observational references are available. These sea ice metrics give a detailed view of sea ice state and highlight major deficiencies in the sea ice

simulation. The SITool is written in the open-source language Python and distributed under the Nucleus for European Modelling of the Ocean (NEMO) standard tools, coming together with the reference code and documentation to make sure the final results are traceable and reproducible.

Here, the SITool version 1.0 (v1.0) is applied to evaluate the performances of bi-polar historical sea ice simulations under the experimental protocols of the CMIP6 Ocean Model Intercomparison Project

(OMIP, Griffies et al., 2016). OMIP provides global ocean-sea ice model simulations with a prescribed atmospheric forcing, which gives the opportunity to intercompare sea ice model performance under fully controlled conditions. In OMIP, two streams of experiments were carried out: OMIP1, forced by the Coordinated Ocean-ice Reference Experiments version 2 interannual forcing (CORE-II, Large and



Yeager, 2009), and OMIP2, forced by the updated Japanese 55-year atmospheric reanalysis (JRA55-do,

Tsujino et al., 2018). The OMIP protocol ensures a close experimental setup among the different models. Models were run with either atmospheric forcing to identify and attribute the influences of changed atmospheric forcings on sea ice characteristics. Tsujino et al. (2020) and Chassignet et al. (2020) evaluated the impact of atmospheric forcing and horizontal resolution on the global ocean–sea ice model simulations based on the experimental protocols of CMIP6 OMIP. Their studies focused on

the evaluation of ocean components from temperature, salinity, mixed layer depth, kinetic energy to circulation changes. For sea ice, they provided diagnostics on the mean ice concentration and thickness, and interannual variability of ice extent. In this paper, we focus on the sea ice in OMIP simulations in a more systematic manner, including more sea ice variables (e.g., ice edge location, snow depth, and ice drift).

This paper is organized as follows. The SITool (v1.0) with the details of sea ice metrics and diagnostics is described in section 2. The CMIP6 OMIP models and observational references are introduced in section 3. In section 4, the application of the SITool (v1.0) to CMIP6 OMIP and the results of the model performance are presented and discussed. Finally, conclusions and discussion are provided in section 5. Appendix A presents some additional sea ice diagnostics. The source code of the SITool (v1.0) used to

assess the model skills is publicly available as shown in the section on "Code and data availability".

## 2 Overview of SITool (v1.0)

A schematic overview of the SITool (v1.0) workflow and its application in evaluating the CMIP6 OMIP model performance is shown in Fig. 1. The methods of the metrics calculation are discussed below followed Massonnet et al. (2011) with some modifications. Namely, more observational references are

used to calculate the typical errors and to do the comparisons, and more sea ice variables (e.g., ice edge location and snow depth) and models are included. The SITool (v1.0) also produces additional sea ice diagnostics to help understand why metrics vary from one dataset to the next. All the sea ice data from model outputs and observational references are interpolated to the polar stereographic 25 km resolution grid to allow point-by-point comparison and to avoid the systematic bias of sea ice extent or area under

different grids, due to differences in land-sea masks.

The general approach to derive metrics is by computing scaled absolute errors. We first compute the errors (in absolute value) between some simulated characteristics (e.g., sea ice extent) in individual models and the corresponding characteristic in observational references, respectively. Then, we scale these errors by a typical error to finally get the corresponding metric. The typical error is defined as the

absolute difference of the relevant characteristic between two observational references, and is therefore a proxy for observation uncertainty. Because our metrics are defined as scaled absolute errors, they are





oriented positively meaning that lower values indicate better skill, and a value of 1 means that model error is comparable to observational uncertainty.

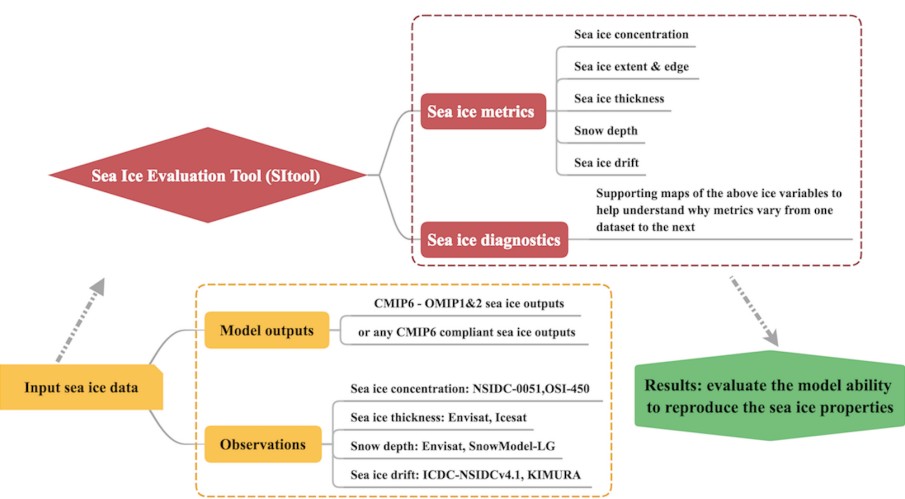

**Figure 1. Schematic overview of the SITool (v1.0) and its application to the CMIP6 OMIP model evaluation.**

The methods to calculate the metrics of ice concentration on the mean state, interannual variability, and trend in both hemispheres are introduced here. For the mean state evaluation, we compute the monthly mean ice concentration over the study period (1980-2007 for the CMIP6 OMIP model evaluation), and calculate the absolute difference between individual models and observational references over 12 months at each grid cell. For the interannual variability and trend evaluation, we compute the standard deviation and linear regression on the monthly anomalies of ice concentration over the study period, and compute the absolute difference between individual models and observational references at each grid cell. Then we average these errors spatially weighted by grid cell areas. The typical errors are the differences between two observational references on the mean state, interannual variability, and trend by applying the same method shown before. The differences between individual models and observational references are computed separately and scaled by corresponding typical errors to get the metrics on ice concentration. The mean February and September ice concentration differences between model outputs and the observational reference, and between two observational references in both hemispheres are provided for diagnosis.

The ice extent is calculated as the total area of grid cells with the ice concentration above 15%. The same procedure is followed for ice extent metrics calculation as for ice concentration, except for the spatial averaging since ice extent is already an integrated quantity. The mean seasonal cycle, monthly anomalies, and trend of ice extent in both hemispheres from different models and two observational





references are provided for diagnosis. The integrated ice-edge error (IIEE) is the total area where the
models and observational references disagree on the ice concentration being above or below 15%
including both the ice extent error and a misplacement error (Goessling et al., 2016). For the mean IIEE
evaluation, we compute the monthly mean IIEEs between individual models and two observational
references, and between the two observational references themselves over the study period. The typical
error is the mean IIEE between two observational references. The differences between individual
models and observational references are computed separately and scaled by the typical error to get the
metric on the ice edge location. The mean seasonal cycle of IIEEs between individual models and the
observational reference, and between two observational references in both hemispheres are provided for
diagnosis.

The same procedure is followed for ice thickness and snow depth metrics calculation as for ice
concentration, except for the spatial averaging with equal weight. The mean February and September
ice thickness and snow depth differences between model outputs and the observational reference, and
between two observational references in both hemispheres can be provided for diagnosis when
observational references are available. For the CMIP6 OMIP model evaluation before 2007, the ice
thickness and snow depth observations are limited to some months. The mean winter (February for the
Arctic and September for the Antarctic) ice thickness and snow depth from ESA's Environmental
Satellite (Envisat) radar altimeter data and the differences between model outputs and Envisat data are
provided for diagnosis in this study.

The ice drift metrics include the evaluation of both the magnitude and direction of ice vectors by
calculating the mean kinetic energy (MKE) and vector correlation of the ice vectors. The MKE is
computed as:

$$MKE = \frac{1}{2}(u^2 + v^2) \,, \tag{1}$$

where u and v are zonal and meridional components of ice drift, respectively. For the MKE evaluation,
we compute the monthly mean MKE over the study period and calculate the absolute difference
between individual models and observational references over the 12 months at each grid cell. Then we
average these errors spatially with equal weight. The typical error is the difference between two
observational references of the MKE by applying the same method discussed before. The differences
between individual models and observational references are computed separately and scaled by the
typical error to get the metric on the ice drift magnitude. The monthly mean ice vectors during the study
period from individual models and observational references are correlated at each grid point by using a
vector correlation measure, which is a generalization of the simple correlation coefficient between two
scalar time series (Holland and Kwok, 2012). The vector correlation coefficient $r^2$ is computed by
following the equations in Crosby et al. (1993), and the correlation coefficient is scaled (by a number 2)





to keep it between 0 and 1 in our study. The $nr^2$ follows the chi-square distribution with four degrees of
freedom, and the correlations are significant at a level of 99% when $nr^2>8$ with samples less than 64
based on the cumulative frequency distributions in Crosby et al. (1993). The significant correlation
coefficients between individual models and observational references, and between two observational
references are provided for diagnosis at each grid cell. Then we average these significant correlation
coefficients spatially with equal weight. The typical correlation coefficient is a spatially averaged
correlation coefficient between two observational references. As higher correlation coefficients indicate
better skill, the typical correlation coefficients are scaled by the correlation coefficients between
individual models and observational references to make it consistent with other metrics (lower values
indicate better skill). The mean February and September MKE differences and the ice-motion vector
correlation coefficients between model outputs and the observational reference, and between two
observational references in both hemispheres are provided for diagnosis.

## 3 Models and observational references

In this study, the SITool (v1.0) is used to evaluate the CMIP6 OMIP model skills in simulating the
historical sea ice properties for both hemispheres. The CMIP6 OMIP models and a set of observational
references providing ice concentration, thickness, snow depth, and ice drift are introduced in this
section. Two sets of observational references for each sea ice variable are used for comparison.

The CMIP6 OMIP models used are shown in Table 1 with model details such as atmospheric forcing,
ocean models, sea ice models, spatial resolution, and related references. A major improvement in
JRA55-do atmospheric forcing relative to the CORE-II forcing is the increased temporal frequency
from 6 to 3 hours and horizontal resolution from 1.875° to 0.5625°. The surface fields of JRA55-do
forcing have been adjusted to match reference datasets based on high-quality satellite observations and
several other atmospheric reanalysis products, as detailed in Tsujino et al. (2018). Nine models were run
with either CORE-II or JRA55-do forcing; five of them were forced by both CORE-II and JRA55-do
reanalysis; out of the four remaining models, one of them was forced by JRA55-do reanalysis only, and
the other three were forced by CORE-II reanalysis only. The CMCC-CM2-HR4 and CMCC-CM2-SR5
models are different in spatial resolution, which provides an opportunity to identify the influence model
resolution on sea ice simulation. The CORE-II forcing dataset has not been updated since 2009 and the
two GFDL models only provide the model outputs until 2007. This is why the evaluation period is
chosen as 1980-2007 for ice concentration, extent, and edge location (the corresponding observations
are available from 1980). The evaluation period is 2003-2007 for ice thickness, snow depth, and ice
drift because some observational references are limited before 2003, and then the corresponding metrics
are only on the mean state.





Table 1. The details of nine CMIP6-OMIP models evaluated in the study.

| Model | Institution | Atmospheric forcing | Ocean Model | Sea Ice Model | Spatial Resolution | References |
|---|---|---|---|---|---|---|
| CMCC-CM2-HR4 | CMCC | JRA55-do | NEMO3.6 | CICE4 | ORCA-0.25° | Cherchi et al. (2019) |
| CMCC-CM2-SR5 | CMCC | CORE-II/JRA55-do | NEMO3.6 | CICE4 | ORCA-1° | |
| EC-Earth3 | EC-Earth | CORE-II/JRA55-do | NEMO3.6 | LIM3 | ORCA-1° | EC-Earth consortium (2019) |
| GFDL-CM4 | NOAA GFDL | CORE-II | OM4 | SIS2 | tripolar, ~0.25° | Held et al. (2019) |
| GFDL-OM4p5B | NOAA GFDL | CORE-II | OM4 | SIS2 | tripolar, ~0.5° | Zadeh et al. (2018) |
| IPSL-CM6A-LR | IPSL | CORE-II | NEMO-OPA | LIM3 | eORCA-1° | Boucher et al. (2020) |
| MIROC6 | JAMSTEC-AORI-NIES-RCCS | CORE-II/JRA55-do | COCO 4.9 | COCO 4.9 | tripolar, ~1°✕(0.5-1)° | Tatebe et al. (2019) |
| MRI-ESM2-0 | MRI | CORE-II/JRA55-do | MRI.COM4.4 | MRI.COM4.4 | tripolar, ~1°✕(0.3-0.5)° | Yukimoto et al. (2019) |
| NorESM2-LM | NorESM | CORE-II/JRA55-do | BLOM | CICE 5.1.2 | tripolar, ~1°✕(0.25-1)° | Seland et al. (2020) |

The observational reference products for sea ice concentration, thickness, snow depth, and ice drift used to compare with model simulations are summarized in Table 2. The first ice concentration product derives from the passive microwave data of the Scanning Multichannel Microwave Radiometer (SMMR), the Special Sensor Microwave Imager (SSM/I), and the Special Sensor Microwave
Imager/Sounder (SSMIS), which are processed by using the NASA Team algorithm (NSIDC-0051, Cavalieri et al., 1996). The other product is based on the same raw data, but uses the EUMETSAT Ocean and Sea Ice Satellite Application Facility algorithm (OSI-450, Lavergne et al., 2019).

Our first ice thickness product is derived from the measurements of ESA's Envisat radar altimeter and
provided by the Centre of Topography of Oceans and Hydrosphere (CTOH, Guerreiro et al., 2017). The other ice thickness product is from the measurements of the NASA's Ice, Cloud, and land Elevation Satellite (ICESat) Geoscience Laser Altimeter System (GLAS), and reprocessed separately for the Arctic (NSIDC-0393, Yi and Zwally, 2009) and Antarctic (Kurtz and Markus, 2012). The sea ice freeboard is less uncertain in observations than thickness, however, only five CMIP6 OMIP models
provide sea ice freeboard and the model's seawater densities, sea ice densities, and snow densities are not provided to calculate the freeboard. The Envisat data includes ice thickness and thickness uncertainties from November to April for the Arctic with coverage up to 81.5° N and May to October for the Antarctic from 2003. The ICESat data used here includes 13 measurement campaigns for the Arctic and 11 for the Antarctic during 2003-2007, and these campaign periods are limited to the months
of February-March, March-April, May-June, October-November with each roughly 33 days. The





comparisons between individual models and the two observational references are thus restricted to these months when data is available. The months chosen for the comparison are different from two ice thickness observational references, which can contribute to the differences in ice thickness performance metrics.


Table 2. Observational references used to compare with model simulations.

| Variable (period) | Data name and references | Available online at: |
|---|---|---|
| Sea ice concentration (1980-2007) | NSIDC-0051: Cavalieri et al. (1996) | https://nsidc.org/data/nsidc-0051 |
| | OSI-450: Lavergne et al. (2019) | http://osisaf.met.no/p/ice/ |
| Sea ice thickness (2003-2007) | Envisat: Guerreiro et al. (2017) | http://ctoh.legos.obs-mip.fr/data/sea-ice-products/sea-ice-thickness |
| | ICESat: NH: Yi and Zwally (2009) SH: Kurtz and Markus (2012) | NH: https://nsidc.org/data/nsidc-0393 SH: https://earth.gsfc.nasa.gov/index.php/cryo/data/antarctic-sea-ice-thickness |
| Snow depth (2003-2007) | Envisat: Guerreiro et al. (2017) | http://ctoh.legos.obs-mip.fr/data/sea-ice-products/sea-ice-thickness |
| | SnowModel-LG: Liston et al. (2020) and Stroeve et al. (2020) | http://dx.doi.org/10.5067/27A0P5M6LZBI |
| Sea ice drift (2003-2007) | ICDC-NSIDCv4.1: Tschudi et al. (2019) | https://icdc.cen.uni-hamburg.de/en/seaicedrift-satobs-global.html |
| | KIMURA dataset: KIMURA et al. (2013) | https://ads.nipr.ac.jp/vishop/ |

The Envisat thickness data also includes snow depth and associated uncertainty. The other snow depth product derives from a Lagrangian snow-evolution model (SnowModel-LG) forced by the European Centre for Medium-Range Weather Forecasts (ECMWF) 5th Generation (ERA5) atmospheric reanalysis, and NSIDC sea ice concentration and trajectory datasets (Liston et al., 2020; Stroeve et al.,

2020). The SnowModel-LG data is only provided for the Arctic Ocean. The SnowModel-LG data used to do the comparison is in the same months as the Envisat data from 2003 to 2007.

The first ice drift product is processed by NSIDC and enhanced by the Integrated Climate Data Center (ICDC-NSIDCv4.1). This product derives from SMMR, SSM/I, SSMIS, and the Advanced Very High Resolution Radiometer (AVHRR) for the Antarctic. In addition to the above data, data of the Advanced

Multichannel Scanning Radiometer-Earth Observing System (AMSR-E), observations of the International Arctic Buoy Program (IABP), and ice drift derived from NCEP/NCAR surface winds are used for the Arctic Ocean. The second ice drift dataset is processed by Kimura et al. (2013) and derived from the AMSR-E data for both hemispheres from 2003.

The ice vectors are reprocessed before calculating the ice drift metrics. The ice vectors from

observational references and models are rotated and interpolated to the polar stereographic grid. The monthly mean ice vectors of the observational references are computed when there are more than 10



days with valid daily drift data. The ICDC-NSIDCv4.1 ice drift data was shown to be biased low (i.e., too slow) relative to buoy data (Schwegmann et al. 2011; Barthélemy et al., 2018) and is therefore corrected by multiplying the drift components with a correction factor of 1.357 (Haumann et al., 2016).

The ice vectors from observational references and models are removed when ice concentrations are below 50%, or the data is closer than 75 km to the coast, or with a spurious value, to reduce the spatial and temporal noise by following Haumann et al. (2016).

## 4. SITool application and results

The SITool (v1.0) described in Section 2 is applied in this section to assess the performance of the sea

ice simulations for both hemispheres carried out under the CMIP6 OMIP1 and OMIP2 protocols. Models forced by CORE-II atmospheric reanalysis data (OMIP1) or JRA55-do reanalysis data (OMIP2) are marked as <model name + /1 or /2>, respectively. The OMIP1 and OMIP2 model means shown below are from five models of CMCC-CM2-SR5, EC-Earth3, MIROC6, MRI-ESM2-0, and NorESM2-LM providing both OMIP1 and OMIP2 model outputs. All the sea ice data from models and

observational references are interpolated to the NSIDC-0051 polar stereographic 25 km resolution grid for comparison. The typical errors are the differences between two observational references for the ice concentration, extent, edge location, and ice drift. The typical errors of ice thickness and snow depth are calculated from the thickness and snow depth uncertainties of specific months from Envisat data because different observational months limit the calculation of differences between two observational

references.

### 4.1 Sea ice concentration, extent, and edge location

Figure 2 shows that model errors on ice concentration simulations are around two to five times the observational uncertainty and the ice concentration simulations are much closer to the NSIDC-0051 data (Fig. 2a) compared to the OSI-450 data (Fig. 2b). In general, the overall ice concentration

simulations (mean state, interannual variability, and trend) in both hemispheres are improved under OMIP2 protocol, forced by JRA55-do reanalysis. This is identified in Fig. 2a by comparing the 5 OMIP1 and OMIP2 model mean values (last two rows), and also by comparing five models' values separately under either OMIP protocol. The overall ice concentration simulations in both hemispheres are also improved in CMCC-CM2-HR4/2 with increased spatial resolution of ocean-sea ice model

compared to CMCC-CM2-SR5/2 (first and third rows). The improved ice concentration simulations are found compared to different observational references except for the interannual variability of the Antarctic ice concentration compared to the OSI-450 data as shown in the fifth column of Fig. 2b.

The metrics on the interannual variability of ice concentration (second and fifth columns) are the highest among all metrics, which indicates relatively lower skill on the simulation of ice concentration



variability in both hemispheres compared to the mean state and trend. The overall best performance on ice concentration simulations including the mean state, interannual variability, and trend is in NorESM2-LM forced by JRA55-do reanalysis for both hemispheres. To help understand the differences in the ice concentration metrics, the 1980-2007 September and February mean ice concentration differences between the OSI-450 and NSIDC-0051 data, and between model outputs and the NSIDC-

0051 data are produced for both hemispheres in the appendix A: Figs. A1-A4.

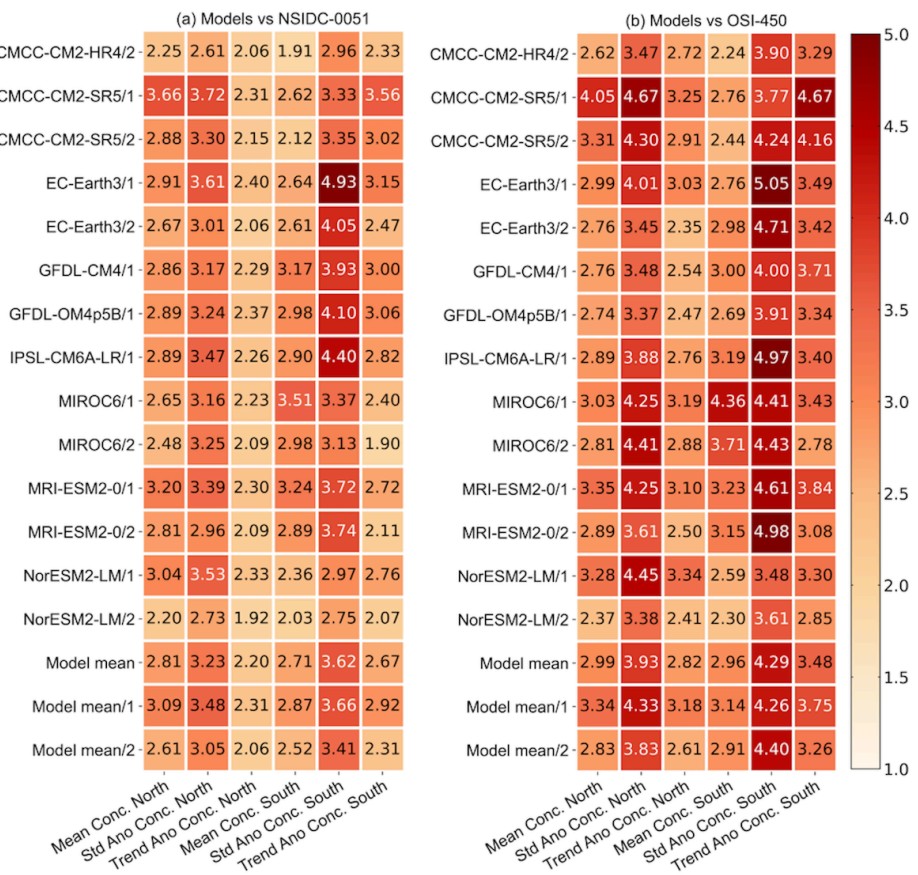

**Figure 2. The ice concentration metrics of 14 model outputs under OMIP1 (/1) and OMIP2 (/2) protocols, 14-model mean (Model mean), 5-OMIP1-model mean (Model mean/1) and 5-OMIP2-model mean (Model mean/2) from CMCC-CM2-SR5, EC-Earth3, MIROC6, MRI-ESM2-0, and NorESM2-LM compared to (a) NSIDC-0051 and (b) OSI-450 data. The six columns correspond to**

**model performance metrics on the mean state, interannual variability, and trend of the Arctic and Antarctic ice concentration during 1980-2007.**

Figure 3 reveals that the monthly ice extent differences between two observational references (observational uncertainty, black × vs. cyan +) are much smaller compared to the model bias in both hemispheres. The negative ice extent biases under OMIP1 protocol in the summer of both hemispheres

are reduced under OMIP2 protocol (Figs. 3a and 3b, red solid vs. dash-dotted) by changing the atmospheric forcing to JRA55-do reanalysis. The reduced mean ice extent biases in the summer under OMIP2 protocol are also identified in Tsujino et al. (2020) (see their Fig. 22 and Table. D7). In the





boreal winter (Fig. 3a), the 5-model mean ice extent under OMIP1 and OMIP2 protocols show no obvious difference (red solid vs. dash-dotted), and the ice extent among most models are close to the

observational references except for the MIROC6 (orange) and MRI-ESM2-0 (gray). In the austral winter (Fig. 3b), large spread exits for the ice extent simulation, and the positive ice extent bias under OMIP1 protocol (red solid) changed to the negative bias under OMIP2 protocol (red dash-dotted) without reduction.

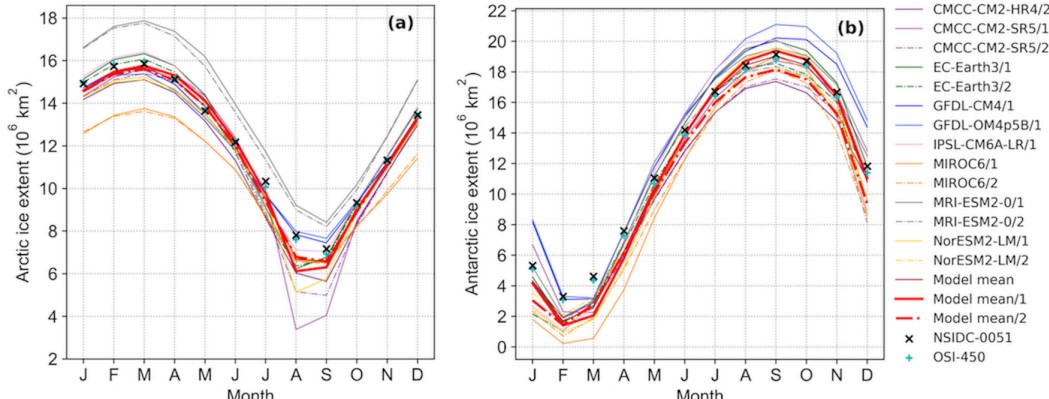

**Figure. 3. The 1980–2007 mean seasonal cycle of ice extent ($10^6$ km$^2$) in the (a) Arctic and (b) Antarctic from 14 model outputs under OMIP1 (/1) and OMIP2 (/2) protocols, 14-model mean (brick red solid), 5-model mean under OMIP1 or OMIP2 protocol (red solid vs. dash-dotted), NSIDC-0051 (black ×) and OSI-450 (cyan +). The 5-model mean is from CMCC-CM2-SR5, EC-Earth3, MIROC6, MRI-ESM2-0, and NorESM2-LM, and the model outputs under OMIP2 protocol from these five models are shown in dash-dotted lines.**

The biases of 5-model mean ice extent monthly anomalies under OMIP1 protocol compared to the observational mean (green vs. black solid) are reduced under OMIP2 protocol (orange vs. black solid) in both hemispheres as shown in Fig. 4. In the Arctic (Figs. 4a and 4b), the negative biases of ice extent monthly anomalies during 1980-1982 and after 1998, as well as positive bias during 1986-1990 are reduced in the OMIP2 model mean (orange vs. green solid). However, the declining trend of ice extent

from the observational mean (black dashed) is close to the OMIP1 model mean (green dashed) but not the OMIP2 model mean (orange dashed). This can be caused by the error compensation of the negative ice extent biases to observational mean during 1980-1982 and after 1998 in the OMIP1 model mean. In the Antarctic (Figs. 4c and 4d), the reduced bias is obvious after 1988 in the OMIP2 model mean (orange vs. green solid). The increasing trend of the Antarctic ice extent in the observational mean

(black dashed) is not shown in the OMIP1 and OMIP2 mean (green and orange dashed). The ice extent monthly anomalies in each model under OMIP1 and OMIP2 protocols are compared separately, and the improvements on the simulations of ice extent interannual variability are found in the OMIP2 model outputs of individual models (not shown). The improved interannual variability of ice extent in the OMIP2 simulations is also identified in Tsujino et al. (2020) (see their Figs. 22 and 23).



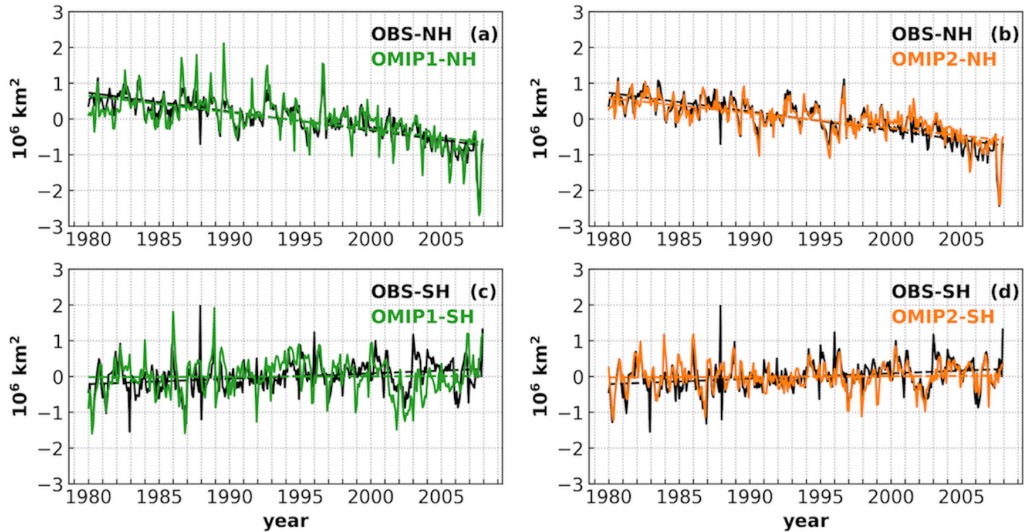


**Fig. 4. The 1980–2007 monthly anomalies of ice extent ($10^6$ km$^2$) from the observational mean of NSIDC-0051 and OSI-450 (black solid), 5-model mean under OMIP1 or OMIP2 protocol (green vs. orange solid) in the Arctic (a, b) and Antarctic (c, d). The 5-model mean is from CMCC-CM2-SR5, EC-Earth3, MIROC6, MRI-ESM2-0, and NorESM2-LM. The dashed lines are the trends computed from linear regression over 1980-2007.**

Figs. 5a and 5b shows that the model errors on ice extent simulation are much larger than the observational uncertainty in most cases, and the large values on the fifth columns are due to the very low typical error ($0.0009 \times 10^6$ km$^2$) of the Antarctic interannual ice extent variability. In general, the ice extent simulations on the mean state and interannual variability for the Arctic, as well as the interannual variability and trend for the Antarctic are improved under OMIP2 protocol, forced by JRA55-do

reanalysis. This is identified in Figs. 5a and 5b by comparing the 5 OMIP1 and OMIP2 model mean values (last two rows), though there are several exceptions for the simulation of individual models under either OMIP protocol. The improved ice extent simulations are identified compared to different observational references.

The simulation of Arctic ice extent trend under OMIP2 protocol is not better than that under OMIP1

protocol (the third columns in Figs. 5a and 5b), which is due to the error compensation of the monthly anomalies biases of the ice extent during different periods under OMIP1 protocol as explained in Figs. 4a and 4b. This error compensation can change the trend and make it close to the observational references even though the monthly anomalies are not well presented in the OMIP1 models. The unimproved Antarctic mean ice extent under OMIP2 protocol can also be found in Fig. 3b where the ice

extent bias in the austral winter is not reduced under OMIP2 protocol. This is not consistent with what we found for the improvement in the ice concentration simulation under OMIP2 protocol, which is possibly because ice extent cancels out regional concentration differences. The overall best performance





on ice extent simulation including the mean state, interannual variability, and trend is in EC-Earth3/1 for the Arctic and in MRI-ESM2-0/2 for the Antarctic.

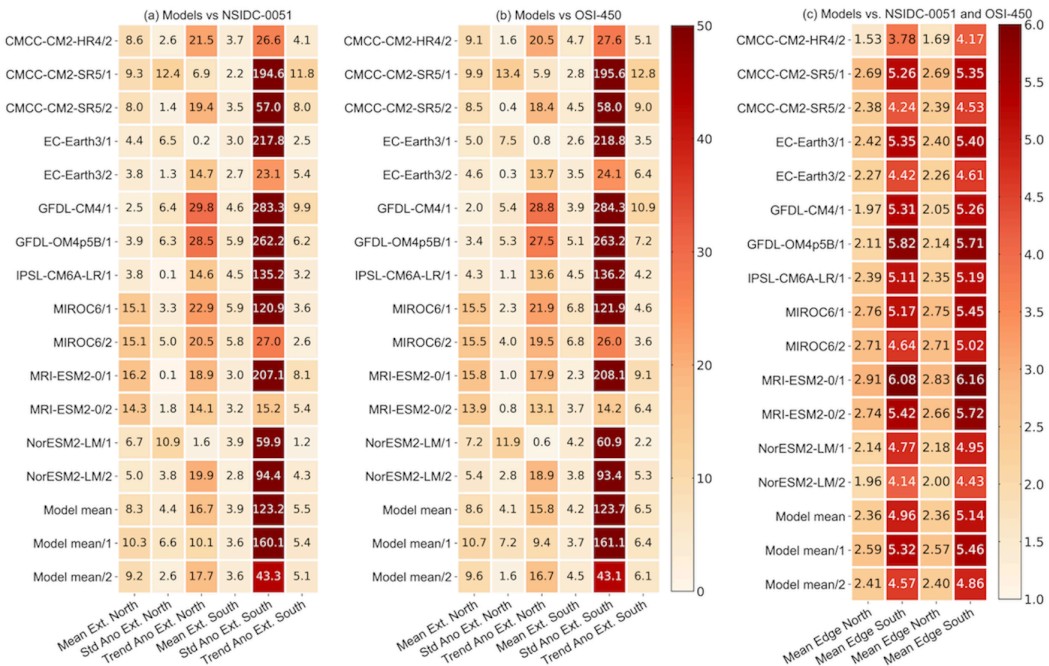


**Figure 5. The ice extent metrics of 14 model outputs under OMIP1 (/1) and OMIP2 (/2) protocols, 14-model mean (Model mean), 5-OMIP1-model mean (Model mean/1) and 5-OMIP2-model mean (Model mean/2) from CMCC-CM2-SR5, EC-Earth3, MIROC6, MRI-ESM2-0, and NorESM2-LM compared to (a) NSIDC-0051 and (b) OSI-450 data. The six columns correspond to model performance metrics on the mean state, interannual variability, and trend of the Arctic and Antarctic ice extent during 1980-2007.**
**(c) The mean state ice edge location metrics during 1980-2007 in both hemispheres compared to the NSIDC-0051 (first two columns) and OSI-450 data (last two columns).**

To gain insights in the spatial distribution of errors, we then apply the IIEE (Goessling et al., 2016) as introduced in section 2. In both hemispheres, the IIEEs between models and NSIDC-0051 are obviously much larger than that between two observational references as shown in Fig. 6. The largest model errors
and model spreads are in the summer of both hemispheres. The IIEE under OMIP1 protocol is much reduced under OMIP2 protocol especially in the summer of both hemispheres (red solid vs. dash-dotted) by changing the atmospheric forcing to JRA55-do reanalysis. In both hemispheres, the large IIEE in CMCC-CM2-SR5/2 (light purple dashed) is reduced in CMCC-CM2-HR4/2 (dark purple solid) with increased spatial resolution of ocean-sea ice model during all the seasons. To identify the ice edge
location errors of various models, the contours of 15% concentration derived from the 1980-2007 September and February mean ice concentration are also shown for both hemispheres in the appendix A: Figs. A1-A4.

The mean state ice edge location metrics in Fig. 5c shows that model errors on ice edge location simulations are around two to six times the observational uncertainty, and the ice edge location



simulations in the Arctic are much better than that in the Antarctic. The mean state ice edge location simulations in both hemispheres are improved under OMIP2 protocol, which is identified in Fig. 5c by comparing the 5-model mean values (last two rows), and also by comparing five models' values separately under either OMIP protocol. The mean state ice edge location simulations in both hemispheres are also improved in CMCC-CM2-HR4/2 with increased ocean-sea ice model resolution

compared to CMCC-CM2-SR5/2 (first and third rows). The improved ice edge location simulations are identified compared to different observational references. The best performance on the mean state ice edge location simulations is in CMCC-CM2-HR4/2 for both hemispheres.

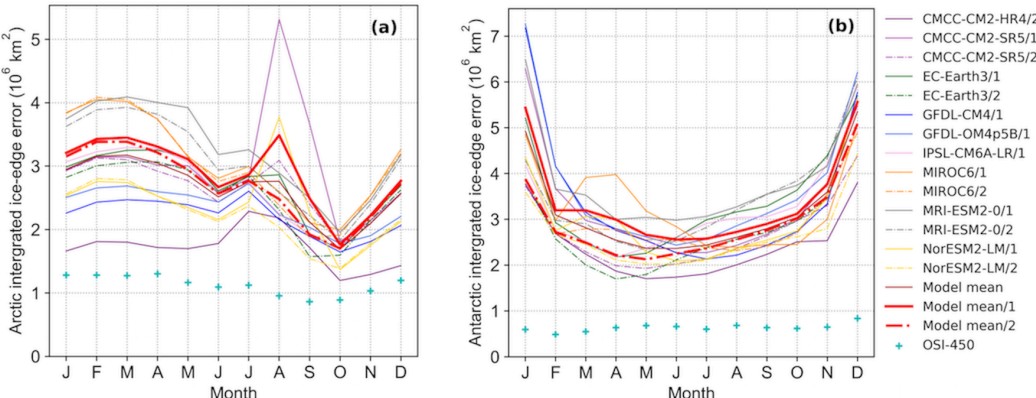

**Figure. 6. The 1980–2007 mean seasonal cycle of the Integrated Ice-Edge Error (IIEE, $10^6$ km$^2$) in the (a) Arctic and (b) Antarctic**
**between 14 model outputs/OSI-450 and NSIDC-0051. The 14-model mean (brick red solid), 5-model mean under OMIP1 or OMIP2 protocol (red solid vs. dash-dotted), and OSI-450 IIEE (cyan +) are marked. The 5-model mean is from CMCC-CM2-SR5, EC-Earth3, MIROC6, MRI-ESM2-0, and NorESM2-LM, and the model outputs under OMIP2 protocol from these five models are shown in dash-dotted lines.**

## 4.2 Sea ice thickness and snow depth

Fig. 7 shows the mean state ice thickness and snow depth metrics, and the interannual variability and trend metrics are not included here because the observational record is too short to make such an assessment (Tilling et al., 2015). The model errors on the mean ice thickness and snow depth simulations are not obviously larger (even smaller in some models) than the observational uncertainty. The mean ice thickness simulation during 2003-2007 is improved in the Antarctic under OMIP2

protocol, forced by JRA55-do reanalysis. This is identified in Fig. 7a by comparing the 5-model mean values (last two rows), and also by comparing the five models' values separately under either OMIP protocol (an exception in NorESM2-LM compared to the Icesat data). The improved Antarctic mean ice thickness simulations are identified compared to different observational references. The best performance on the mean ice thickness simulation is in IPSL-CM6A-LR/1 for the Arctic, while for the

Antarctic the best performance is in CMCC-CM2-HR4/2 compared to the Envisat data and in GFDL-OM4p5B/1 compared to the ICESat data. The different model performance on the mean ice thickness





simulations by comparing to two observational references are due to the different months chosen for the ice thickness comparison.

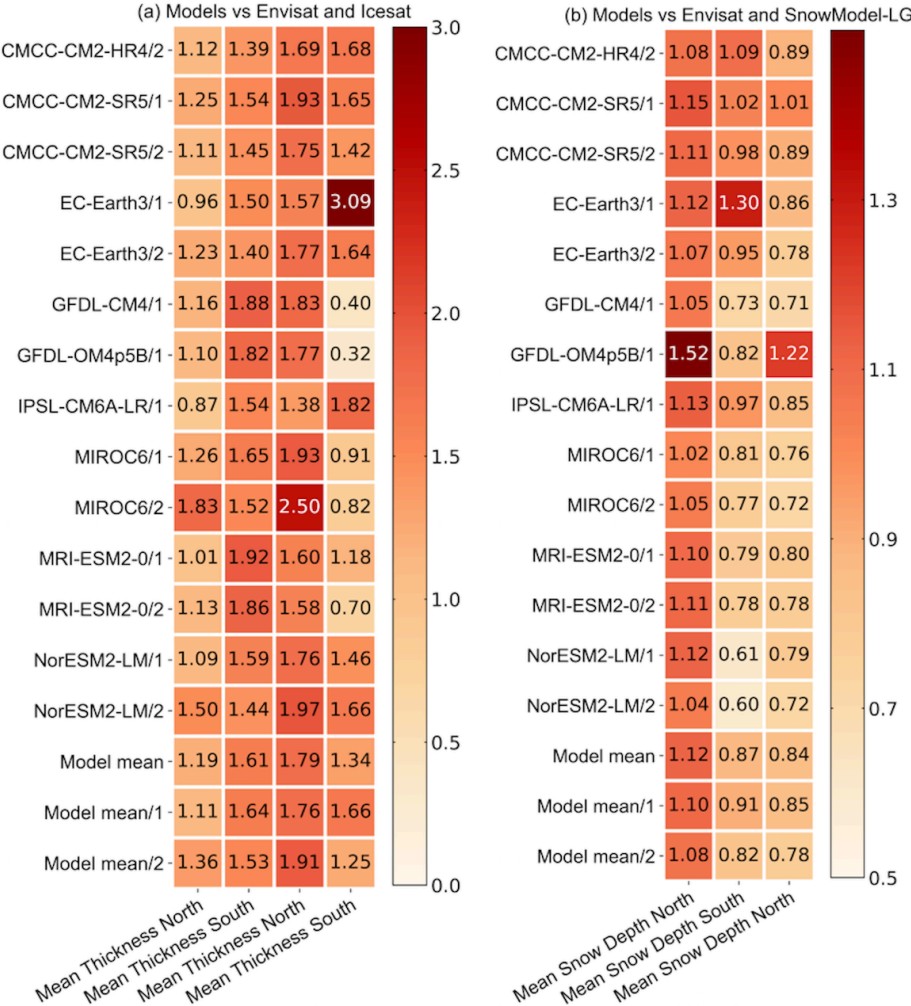

**Figure 7.** The mean state ice thickness (a) and snow depth (b) metrics during 2003-2007 of 14 model outputs under OMIP1 (/1) and OMIP2 (/2) protocols, 14-model mean (Model mean), 5-OMIP1-model mean (Model mean/1) and 5-OMIP2-model mean (Model mean/2) from CMCC-CM2-SR5, EC-Earth3, MIROC6, MRI-ESM2-0, and NorESM2-LM. The four columns in (a) correspond to ice thickness metrics in both hemispheres compared to the Envisat (first two) and ICESat data (last two), and the three columns in (b) correspond to snow depth metrics compared to the Envisat data in both hemispheres (first two) and the SnowModel-LG data in the Arctic (last one).

The mean snow depth simulation during 2003-2007 in both hemispheres improved a bit under OMIP2 protocol, which can be found by comparing 5-model mean values under either OMIP protocol (last two rows) in Fig. 7b. The improvement on the mean snow depth simulation is relatively small compared to other ice metrics. The best performance on the mean snow depth simulation for the Arctic is in MIROC6/1 compared to the Envisat data and in GFDL-CM4/1 compared to the SnowModel-LG data, and for the Antarctic, the best performance is in NorESM2-LM/2 (Fig. 7b). To help understand the





differences in the ice thickness and snow depth metrics, the 2003-2007 winter-mean ice thickness and snow depth from Envisat data, and the differences between model outputs and Envisat data are produced for both hemispheres in the appendix A: Figs. A5-A8.

### 4.3 Sea ice drift


The magnitude and direction of simulated ice drifts are evaluated by calculating the MKE and the vector correlation of the ice vectors. The vector correlation coefficients are measured by using a generalization of the simple correlation coefficient between two scalar time series as introduced in section 2. The significant correlation coefficient at a level of 99% between ICDC-NSIDCv4.1 and

KIMURA data, and between 14 model outputs and KIMURA data in the Arctic (Fig. 8) and Antarctic (Fig. 9) are displayed. The correlation coefficients are much lower between model outputs and the KIMURA data than that between two observational references. This is obvious for the coastal regions of Greenland and Canadian archipelago in the Arctic and the Weddell Sea and the Ross Sea in the Antarctic, as well as the ice edge location of the Weddell Sea among some models. This implies that

model errors on the ice-vector direction simulations are much larger than the observational uncertainty. The correlation coefficients are higher under OMIP2 protocol than that under OMIP1 protocol (third vs. second column in Figs. 8 and 9), which indicate the improvement on the ice-vector direction simulation when forced by JRA55-do atmospheric forcing in both hemispheres.

Figure 10 shows that model errors on the mean ice drift simulations are larger than the observational

uncertainty. In general, the ice drift simulations on the magnitude (Fig. 10a) and direction (Fig. 10b) in both hemispheres are improved under OMIP2 protocol, forced by JRA55-do reanalysis. This is identified from the 5 OMIP1 and OMIP2 model mean values (last two rows), and also by comparing five models' values separately under either OMIP protocol (an exception in CMCC-CM2-SR5 of the Arctic ice-vector magnitude in Fig. 10a). The improved mean ice drift simulations are identified

compared to different observational references. The overall best performance on sea ice drift simulations including the magnitude and direction is in MIROC6/2 for both hemispheres. To help understand the differences in the ice-motion magnitude metrics, the 2003-2007 September and February mean ice-motion MKE differences between the ICDC-NSIDCv4.1 and KIMURA data, and between model outputs and the KIMURA data are produced for both hemispheres in the appendix A: Figs. A9-

A12.



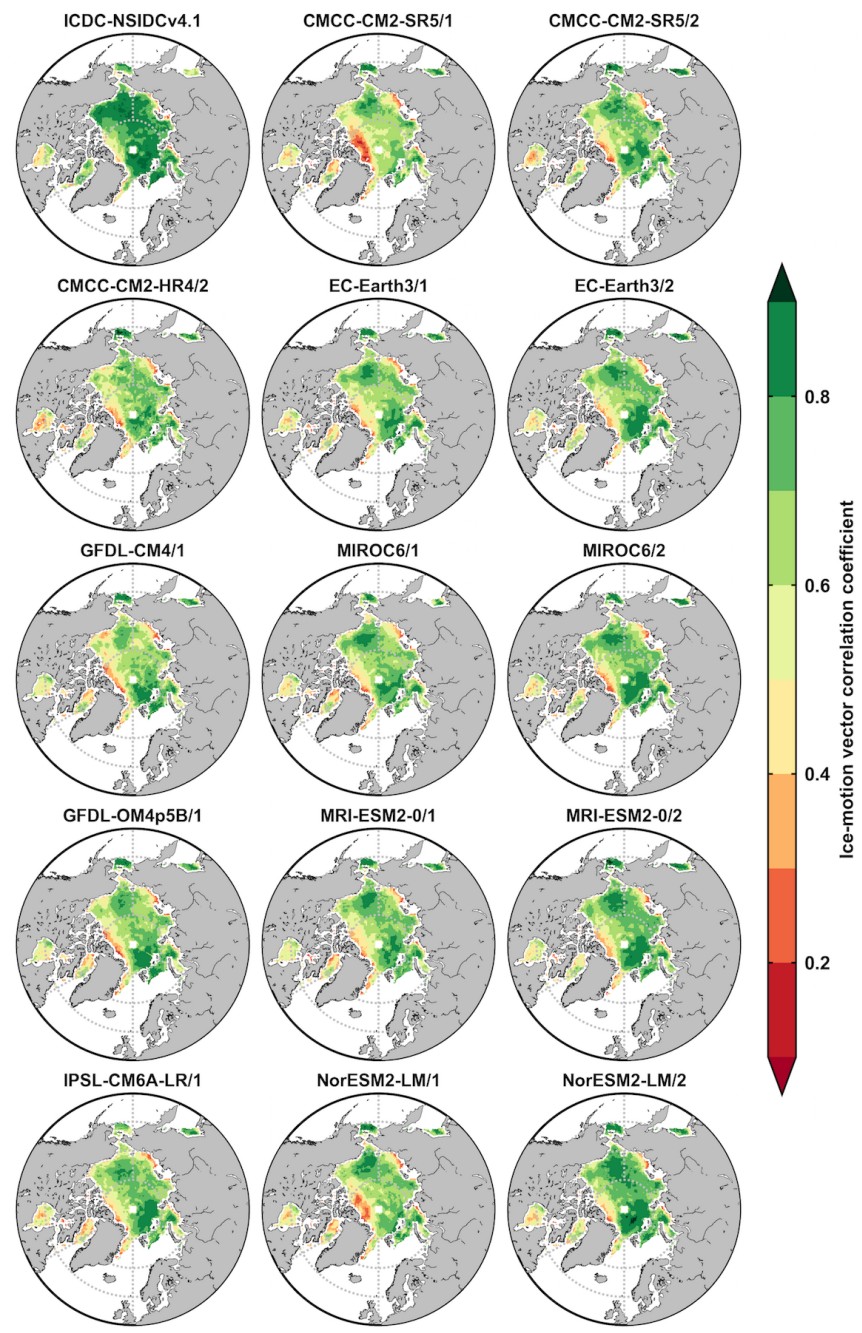

**Figure 8. The significant Arctic ice-motion vector correlation coefficients during 2003–2007 at a level of 99% between ICDC-NSIDCv4.1/model outputs and the KIMURA data ($m^2 \ s^{-2}$). The second and third columns are from 5 OMIP1 and OMIP2 model outputs of CMCC-CM2-SR5, EC-Earth3, MIROC6, MRI-ESM2-0, NorESM2-LM, respectively.**


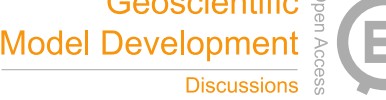

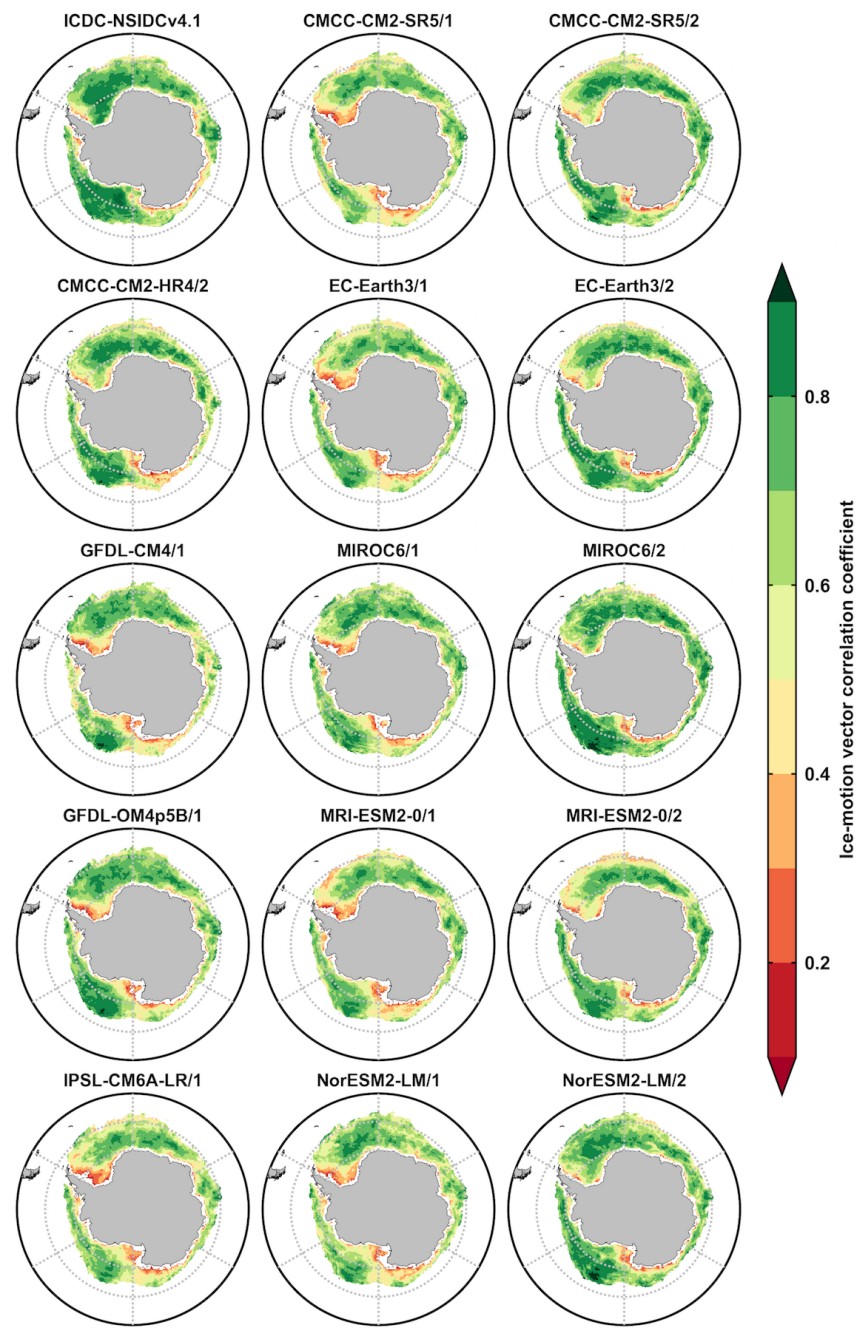

**Figure 9.** The significant Antarctic ice-motion vector correlation coefficients during 2003–2007 at a level of 99% between ICDC-NSIDCv4.1/model outputs and the KIMURA data ($m^2$ $s^{-2}$). The second and third columns are from 5 OMIP1 and OMIP2 model outputs of CMCC-CM2-SR5, EC-Earth3, MIROC6, MRI-ESM2-0, NorESM2-LM, respectively.


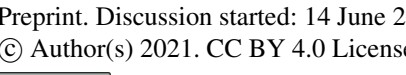



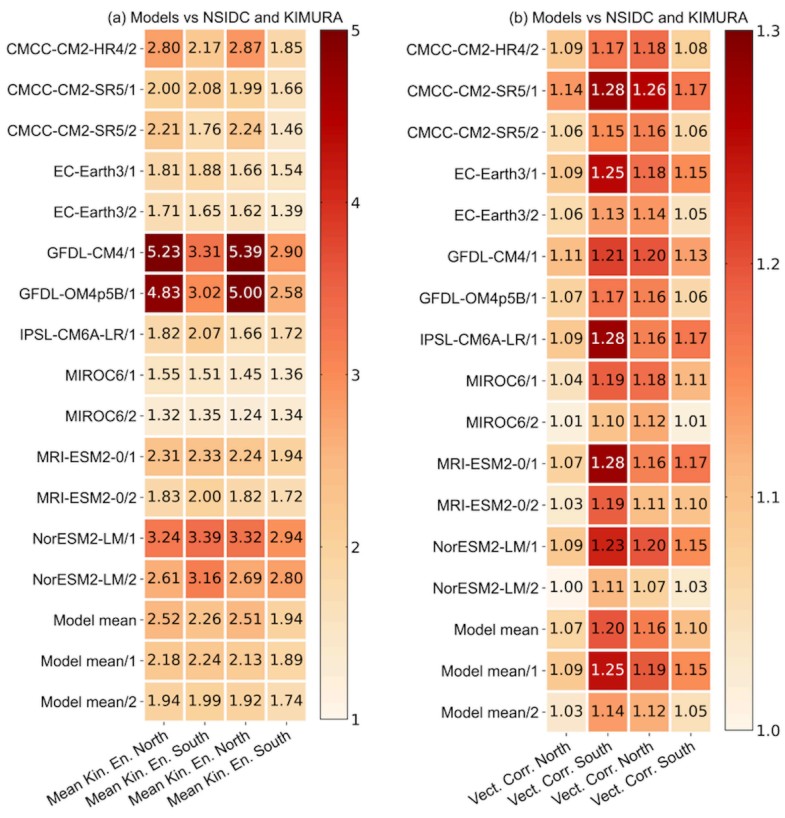

**Figure. 10.** The ice drift metrics of 14 model outputs under OMIP1 (/1) and OMIP2 (/2) protocols, 14-model mean (Model mean), 5-OMIP1-model mean (Model mean/1) and 5-OMIP2-model mean (Model mean/2) from CMCC-CM2-SR5, EC-Earth3, MIROC6, MRI-ESM2-0, and NorESM2-LM. The four columns correspond to model performance metrics on the (a) mean kinetic energy (MKE) and (b) the vector correlations during 2003-2007 of the Arctic and Antarctic compared to the ICDC-NSIDCv4.1 (first two) and KIMURA data (last two).

## 5 Conclusions and discussion

The SITool (v1.0), a performance metrics and diagnostics tool for CMIP6-compliant sea ice outputs, is introduced in this paper. The evaluation includes ice concentration, extent, edge location, thickness, snow depth, and ice drift. The SITool (v1.0) provides rating scores for each sea ice variable in both hemispheres by comparing them to a set of observational references, using at least two such observational references to account for the role of observational uncertainty in the evaluation process. In this paper, we evaluate the CMIP6 OMIP sea ice simulations with SITool (v1.0) to demonstrate the proof of concept and potentialities behind it. Specifically, we evaluate the performances of OMIP historical sea ice simulations (1980-2007 for sea ice areal properties, 2003-2007 for ice drift, thickness and snow depth).

Our main findings on CMIP6 OMIP simulations are summarized below. By changing the atmospheric forcing from CORE-II to JRA55-do reanalysis data, the improvement are identified in (1) the ice





concentration simulations including the mean, interannual variability and trend in both hemispheres, (2)
the ice extent simulations including the mean and interannual variability in the Arctic, as well as the
interannual variability and trend in the Antarctic, (3) the mean ice edge location simulations in both
hemispheres, (4) the mean ice thickness simulations in the Antarctic and the mean snow depth
simulations in both hemispheres, (5) the ice drift simulations including the magnitude and direction in
both hemispheres. By increasing the horizontal resolution of CMCC-CM2 ocean-sea ice model, the
improvements are identified in the sea ice concentration (mean, interannual variability and trend) and
the mean ice edge location simulations in both hemispheres.

In general, model errors are larger than observational uncertainty, and model performances on the ice
concentration, extent, edge location, and ice drift simulations are consistent when comparing to
different observational references. For the ice thickness and snow depth evaluation, the rating scores are
not consistent compared to different observational references, which is due to the limited observations
and to the fact that different months were chosen for comparison during 2003-2007. This finding shows
that sea ice thickness and snow depth estimates are still at a more early stage of maturity compared to
datasets of sea ice concentration or drift. The metrics of each ice variable are ranked for the cross-metric
analysis, and there is no link between the performance in one ice variable and the performance in
another. For example, for the five models (spatial resolution of 1 degree) forced by JRA55-do
reanalysis, the NorESM2-LM/2 is the best on the ice areal simulation, but in the middle for the ice
thickness and the worst for the ice-motion magnitude simulation, and the MIROC6/2 is the best on the
ice drift simulation, but the worst for the ice thickness simulation.

The improvements of mean ice concentration simulations in the summer for both hemispheres by
changing the atmospheric forcing and increasing the horizontal resolution are also identified in Tsujino
et al. (2020) and Chassignet et al. (2020). The reduced mean ice extent bias in boreal summer and much
improved interannual variability of ice extent in OMIP2 simulations are also proved in Tsujino et al.
(2020). For the mean ice thickness simulation, Chassignet et al. (2020) also shows that the improvement
is not obvious by increasing the horizontal resolution of ocean-sea ice models. To understand the
processes leading to the improvement of model simulations under different atmospheric forcings and
ocean-sea ice models, we will discuss the sensitivity of sea ice simulation to CMIP6 OMIP model
physics in an upcoming companion paper.

The metrics make a summary of the model performance on different aspects of the sea ice system to
help detect the inter-model differences or track the time-evolution of model performance efficiently.
However, the usage of metrics comes at the risk of over-interpretation by summarizing all the complex
behavior of models to one number. In fact, a good metric can be obtained for many wrong reasons, so
that we do not recommend relying exclusively on these metrics to orient strategic choices regarding,





e.g., sea ice model development. While it is running, SITool (v1.0) produces ancillary maps and time series that can be consulted by the expert to understand the origin of one particular metric value.

While SITool (v1.0) is primarily designed to assess ocean-sea ice simulations forced by atmospheric reanalysis, it can also be used to evaluate coupled model simulations (e.g., CMIP6 historical runs). We draw the reader's attention to the fact that, in that case, several metrics may become less relevant and less easy to interpret. Indeed, a coupled model is not supposed to produce sea ice output that is in phase with real observations due to the presence of irreducible climate internal variability. This is particularly

true for the evaluation of sea ice thickness and snow depth, for which the limited time span (2003-2007) is likely not enough to draw robust conclusions regarding model performance.

**Appendices A: Sea ice diagnostics**

In this appendix, additional sea ice diagnostics are given to help understand why metrics vary from one dataset to the next. The spatial distribution of the differences between model simulations and the

observational reference is presented Figs. A1-A12 and the model simulations under OMIP1 and OMIP2 protocols are listed in the second and third columns, respectively. This includes the 1980-2007 September and February mean ice concentration differences (Figs. A1-A4), the 2003-2007 winter mean ice thickness (Figs. A5-A6) and snow depth differences (Figs. A7-A8) (February for the Arctic and September for the Antarctic), and the 2003-2007 September and February mean ice-motion MKE

differences (Figs. A9-A12).





**Figure A1. The 1980-2007 September mean Arctic ice concentration differences between OSI-450/model outputs and the NSIDC-0051 data (colors), and contours of 15% concentration of the NSIDC-0051 data (green lines) and OSI-450/model outputs (magenta lines). The second and third columns are from 5 OMIP1 and OMIP2 model outputs of CMCC-CM2-SR5, EC-Earth3, MIROC6, MRI-ESM2-0, NorESM2-LM, respectively.**




**Figure A2. The 1980-2007 February mean Arctic ice concentration differences between OSI-450/model outputs and the NSIDC-0051 data (colors), and contours of 15% concentration of the NSIDC-0051 data (green lines) and OSI-450/model outputs (magenta lines). The second and third columns are from 5 OMIP1 and OMIP2 model outputs of CMCC-CM2-SR5, EC-Earth3, MIROC6, MRI-ESM2-0, NorESM2-LM, respectively.**





**Figure A3. The 1980-2007 February mean Antarctic ice concentration differences between OSI-450/model outputs and the NSIDC-0051 data (colors), and contours of 15% concentration of the NSIDC-0051 data (green lines) and OSI-450/model outputs (magenta lines). The second and third columns are from 5 OMIP1 and OMIP2 model outputs of CMCC-CM2-SR5, EC-Earth3, MIROC6, MRI-ESM2-0, NorESM2-LM, respectively.**





**Figure A4.** The 1980-2007 September mean Antarctic ice concentration differences between OSI-450/model outputs and the NSIDC-0051 data (colors), and contours of 15% concentration of the NSIDC-0051 data (green lines) and OSI-450/model outputs (magenta lines). The second and third columns are from 5 OMIP1 and OMIP2 model outputs of CMCC-CM2-SR5, EC-Earth3, 525 MIROC6, MRI-ESM2-0, NorESM2-LM, respectively.





**Figure A5. The 2003-2007 February mean Arctic ice thickness from Envisat data (first picture, m) and ice thickness differences between model outputs and Envisat data (m). The second and third columns are from 5 OMIP1 and OMIP2 model outputs of CMCC-CM2-SR5, EC-Earth3, MIROC6, MRI-ESM2-0, NorESM2-LM, respectively.**


**Figure A6.** The 2003-2007 September mean Antarctic ice thickness from Envisat data (first picture, m) and ice thickness differences between model outputs and Envisat data (m). The second and third columns are from 5 OMIP1 and OMIP2 model outputs of CMCC-CM2-SR5, EC-Earth3, MIROC6, MRI-ESM2-0, NorESM2-LM, respectively.



**Figure A7. The 2003-2007 February mean Arctic snow depth from Envisat data (first picture, m) and snow depth differences between model outputs and Envisat data (m). The second and third columns are from 5 OMIP1 and OMIP2 model outputs of CMCC-CM2-SR5, EC-Earth3, MIROC6, MRI-ESM2-0, NorESM2-LM, respectively.**





**Figure A8. The 2003-2007 September mean Antarctic snow depth from Envisat data (first picture, m) and snow depth differences**
**between model outputs and Envisat data (m). The second and third columns are from 5 OMIP1 and OMIP2 model outputs of**
**CMCC-CM2-SR5, EC-Earth3, MIROC6, MRI-ESM2-0, NorESM2-LM, respectively.**



**Figure A9. The 2003-2007 September mean Arctic ice-motion MKE differences between ICDC-NSIDCv4.1/model outputs and the KIMURA data (m² s⁻²). The second and third columns are from 5 OMIP1 and OMIP2 model outputs of CMCC-CM2-SR5, EC-Earth3, MIROC6, MRI-ESM2-0, NorESM2-LM, respectively.**




**Figure A10.** The 2003-2007 February mean Arctic ice-motion MKE differences between ICDC-NSIDCv4.1/model outputs and the KIMURA data ($m^2$ $s^{-2}$). The second and third columns are from 5 OMIP1 and OMIP2 model outputs of CMCC-CM2-SR5, EC-Earth3, MIROC6, MRI-ESM2-0, NorESM2-LM, respectively.




**Figure A11. The 2003-2007 February mean Antarctic MKE differences between ICDC-NSIDCv4.1/model outputs and the KIMURA data (m² s⁻²). The second and third columns are from 5 OMIP1 and OMIP2 model outputs of CMCC-CM2-SR5, EC-Earth3, MIROC6, MRI-ESM2-0, NorESM2-LM, respectively.**







**Figure A12. The 2003-2007 September mean Antarctic MKE differences between ICDC-NSIDCv4.1/model outputs and the KIMURA data (m$^2$ s$^{-2}$). The second and third columns are from 5 OMIP1 and OMIP2 model outputs of CMCC-CM2-SR5, EC-Earth3, MIROC6, MRI-ESM2-0, NorESM2-LM, respectively.**





*Code and data availability.* The latest release of SITool (v1.0) is publicly available on Zenodo at https://doi.org/10.5281/zenodo.4621147 (Lin et al., 2021). The source code of the SITool (v1.0) is developed fully based on freely available Python packages and libraries and is released on the GitHub repository available at https://github.com/XiaLinUCL/Sea-Ice-Evaluation-Tool. CMIP6 OMIP data are freely available from the Earth System Grid Federation. Observational references used in this paper are
detailed in section 3 and listed in Table 2, and they are not distributed with SITool (v1.0) because SITool (v1.0) is restricted to the code as open-source software.

*Author contributions.* All authors contributed to the design and discussion of the study. XL performed the analysis and prepared the SITool codes with the help of FM and TF. XL led the writing of the paper and all authors contributed to the editing of the manuscript.

*Competing interests.* The authors declare that they have no conflict of interest.

*Acknowledgments.* This work has been carried out as part of the Copernicus Marine Environment Monitoring Service (CMEMS) SI3 project. CMEMS is implemented by Mercator Ocean International in the framework of a delegation agreement with the European Union. Xia Lin also received support from the National Natural Science Foundation of China (Grant Nos. 41906190, 41941007 and
41876220). François Massonnet is a F.R.S.-FNRS Research Fellow.

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
