# Peer review of "SITool (v1.0) - a new evaluation tool for large-scale sea ice simulations: application to CMIP6 OMIP"

_Geoscientific Model Development, 2021_

## Author Comment (AC1)

Dear Dr. Robel,

Your reference: GMD-2021-99

We thank all three reviewers for their constructive comments on the earlier version of the manuscript. We have revised our manuscript following their comments and our response is as follows.

**Reviewer 1:**

**General comments**

This manuscript describes the creation and application of a new Python-based software, called "SITool", for evaluating Arctic and Antarctic sea ice in global climate models. The authors utilize SITool to analyze models from the CMIP6 OMIP in terms of their sea ice concentration, thickness, snow depth and ice drift. The authors find that model biases exceed observational uncertainties and note improved model performance using the JRA-55 atmospheric forcing versus the CORE-II atmospheric forcing. No single model performs best in all metrics as there is no link found between performance in one variable and performance in another.

This manuscript is thorough and well-organized. The figures and tables support the discussion well and the analysis clearly demonstrates the utility of SITool. My main comment is that, while discussion of model ranking is distributed through the manuscript, there is no section devoted to synthesizing the cross-metric analysis or a figure documenting the rankings (mentioned in the Conclusions on Page 20, Line 468). I recommend the addition of a short paragraph to the Results section summarizing the findings and implications of the cross-metric analysis. While model ranking may not be the primary goal of the tool, it is mentioned enough in the manuscript to warrant further discussion and context prior to the Conclusions. The authors may consider moving the text on Page 20, Lines 470-473 to the added paragraph and/or including a table of the best and worst performing models for each metric to the main text or Appendix.

Overall I recommend minor revisions to address the cross-metric analysis and the specific comments below.

Thanks for your suggestion. We have added one section 4.4 in lines 497-519 to summarize the findings and implications of the cross-metric analysis. The text on page 20, lines 470-473 of the previous version is added to section 4.4. Table 4 is included in section 4.4 to show the best and worst performing models for six sea ice variables in six models forced by the JRA55-do reanalysis.

**Specific Comments**

Page 1, Line 11: I recommend replacing the phrase "bi-polar" with "Arctic and Antarctic" throughout the manuscript for clarity.

We have checked and replaced this phrase.

Page 2, Line 49: I recommend expanding briefly on what is meant by "rather limited" to describe which sea ice diagnostics are provided in ESMValTool and which are unique to SITool.

Thanks for pointing this out. We have rephrased the sentence in lines 49-50 to describe the sea ice diagnostics in ESMValTool. It is an efficient tool to obtain a broad view on the overall performance of a climate model, and it provides sea ice diagnostics on ice concentration and extent, as well as relationships between sea ice variables. The uniqueness of SITool is summarized in lines 50-52. The Sea Ice Evaluation Tool (SITool) introduced in this paper provides systematic sea ice metrics for assessing large-scale sea ice simulations from various aspects in addition to sea ice diagnostics. More details on the SITool are shown in the next paragraph.

Page 2, Line 52: Can you please clarify what is meant by SITool providing "qualitative" information? The tool seems to be used primarily for calculating model biases and related metrics, which I would consider primarily quantitative.

Yes, you are right. The word 'qualitatively' has been deleted then.

Page 3, Line 92: Can you please clarify here if the interpolation is a component of the SITool workflow or is a preprocessing step that needs to be completed before using SITool?

This is a component of the SITool workflow. We have added this information in line 110.

Page 4, Line 117: It would be helpful to have a brief sentence explaining why February and September were chosen (for example, why February instead of March).

Thanks for your suggestion. We have added one sentence to explain this in lines 152-154. They are the representative months of the summer and winter. In Figs. 3a and 3b, the maximum (minimum) sea ice extent is in September (February) for the Antarctic, and the maximum (minimum) sea ice extent is in March (September) for the Arctic from observations (black × and cyan +). The sea ice extent in February and March are very close for the Arctic. We then choose to show the monthly mean of February and September.

Page 6, Line 184: Please list here the respective resolutions of CMCC-CM2-HR4 and CMCC-CM2-SR5 or provide a reference to Table 1.

The resolutions of CMCC-CM2-HR4 and CMCC-CM2-SR5 are listed in lines 224-225.

Page 9, Line 259 (and page 14, line 359): Throughout the manuscript, I recommend using "finer" or "higher" spatial resolution versus "increased".

Thanks for pointing this out. We have checked this word in the manuscript and changed it into 'higher'.

Page 10, Figure 2 (as well as Figures 5, 7, 10): It would be helpful to remind the reader in each of these figure captions that lower values indicate better skill.

Thanks for your suggestion. We have added the sentence 'Lower values indicate better skill' in the figure captions.

Page 14, Line 355: "…the ice edge location simulations in the Arctic are much better than that in the Antarctic." This is an interesting and logical point that you've quantified. Perhaps this has also been shown elsewhere? If so, reference(s) would be helpful.

We have added two references to discuss the differences between the Arctic and Antarctic ice edge location simulations in lines 412-416. Zampieri et al. (2019) show that the prediction skill of sea ice edge location is 30% lower in the Antarctic than in the Arctic from coupled subseasonal forecast systems. The lower prediction skill in the Antarctic can be related to more complicated ocean dynamic processes there, which decrease the persistence of ice area anomalies (Ordoñez et al., 2018).

Page 16, Line 420: Can you please clarify what is meant by "different observational references" in this sentence? Different from what?

Indeed, this sentence was unclear. We have rewritten it to 'The improved mean ice drift simulations under OMIP2 protocol are found compared to not only the ICDC-NSIDCv4.1 data but also the KIMURA data.' in lines 475-476.

Page 17, Figure 8 (and page 18, Figure 9): I recommend a new color map for these figures as the chosen color map may present challenges for readers with red-green color blindness.

Thanks for pointing this out. We have used a new color map for readers with red-green color blindness.

Page 19, Line 446: On page 6, line 144 the authors write that two observational references are used for each variable, but here the phrase "at least two" is used. Can you please clarify if you mean that SITool is equipped to handle more than two sets of observational references?

More than two observational references can be added to the toolbox. In the SITool (v1.0), two sets of observational references are used for each variable. Then we modified the text here to 'two observational references'.

Page 21, Line 488: "While it is running, SITool (v1.0) produces ancillary maps and time series that can be consulted by the expert to understand the origin of one particular metric value." I believe this means that SITool automatically creates the kinds of maps provided in Appendix A, and if that's true, please reference Appendix A here. It would also be useful to note in Section 2 that SITool automatically outputs the differences (which may be just as useful to some users) in addition to the scaled metrics.

Thanks for your suggestion. We have added this information in lines 560-561. While it is running, SITool (v1.0) produces ancillary maps (Figs. 8, 9 and Figs. A1-A12 in Appendix A) and time series diagrams (Figs. 3, 4, 6). In section 2, we mentioned that in lines 104-106.

**Technical corrections**

Page 2, Line 60: I recommend rephrasing the grammar of the final sentence to something such as:

"The SITool is written in the open-source language Python and distributed under the Nucleus for European Modelling of the Ocean (NEMO) standard tools. SITool is provided with the reference code and documentation to make sure the final results are traceable and reproducible."

Thanks for pointing this out. We have rephrased this sentence in lines 61-63.

**Reviewer 2:**

The manuscript "SITool (v1.0) – a new evaluation tool for large-scale sea ice simulations: application to CMIP6 OMIP" describes a new Python diagnostic tool to evaluate sea ice models in the Arctic and Antarctic over the historical period. Although it is designed primarily for atmospheric reanalysis-forced simulations, as presented in the manuscript, it could be useful in other model frameworks as well. This tool is complementary to other climate model evaluation tools, such as ESMValTool. Comparison with multiple observational datasets allow for evaluation of sea ice concentration, extent, edge location, ice thickness, snow depth, and ice drift. The evaluation of Ocean Model Intercomparison Project runs are used here as example, but also provide results and sea ice model performance.

This manuscript is well within the scope of the journal, as it introduces a novel new tool for evaluating climate model performance on a critical component: sea ice. Consistent, repeatable methods of evaluation like this are greatly needed by the community. It also provides novel results on the impact of the atmospheric forcings on the modeled sea ice (especially that model biases are significantly reduced by using

JRA-55). The title and abstract well capture the key points. Methods are generally clearly described, and the code is well-documented and easily accessible. The paper is generally well structured and clearly written, but some figures could be improved for easier interpretation. It would be helpful to be clearer about how the presented results connect with the code and outputs in the published package. I believe with minor suggested edits and demonstration of code implementation (either by an additional reviewer or by the author, within the repository), this manuscript warrants publishing. Note: this reviewer did not complete a test of the scripts, and it may be useful for this code to be checked and tested by someone who is experienced in working with these output types.

Thanks for your suggestions. The figures are improved by following your suggestions mentioned below. The connection between diagnostic fields, the output results and the codes in the published package are described in Table 1. We have provided the completed code run and the corresponding figures in the repository on Github.

Comments:

- L23-30: [suggestion] Separate out into 4 sentences for improved readability

Thanks for pointing this out. We have separated this sentence in lines 23-30.

- L70: "either" feels imprecise here. Is it accurate to say "both atmospheric forcings, when possible, …"?

Thanks for pointing this out. We have rephrased it in lines 72-73.

- L76: It would be helpful to more clearly state how the diagnostics of concentration and thickness in previous studies (2020) are the same or different from the diagnostics proposed here.

Thanks for your suggestion. More details are added in lines 78-89 to show the diagnostics from previous studies and to show the differences from our study.

- L88-89: Including some sort of table summarizing the diagnostics available, in terms of variable and type (spatial map, mean, IIEE) would be helpful.

This would be very helpful for readers to follow our diagnostics. We have added the Table 1 to summarize the diagnostic fields along with input variables, output results and corresponding figures in this paper, python scripts in the repository, and comments.

- L90: Would be preferable for this to be a complete list of add'l sea ice variables, and then "e.g." is not needed

We have modified this sentence in lines 103-104.

- L92-95: Please discuss somewhere (results or conclusions) the possible implications of interpolation.

Thanks for pointing this out. The differences between the results on the original grid and those after the interpolation are very small. The interpolation yields less than 5% error for each sea ice variable, which indicates that the results are not sensitive to the interpolation method used here. We have added this in lines 111-113.

- L105/Fig. 1: Can this be re-oriented such that the order progresses downwards? (So, sea ice input data above SItool?

Yes, we have revised the figure to put the sea ice input data above SITool.

- L105/Fig. 1: In the "Observations" portion, it would be helpful to make it clearer that extent and edge are also coming from the concentration products. In other words, it would be helpful to be explicit what observations each of the defined "metrics" are compared to

Thanks for pointing this out. We have added in the part of 'Sea ice metrics' that the ice extent and edge metrics are derived from the ice concentration data, which means that they are compared to the observations of NSIDC-0051 and OSI-450 data.

- L106: [style suggestion] I think more subsections to separate out the methods would be helpful. This will be helpful in providing a quick reference for users of SITool.

We have divided this part into three subsections. Thanks for pointing it out.

- L117: This sentence is hard to understand. Perhaps it could be clarified by separating into 2 sentences.

We have modified this sentence in lines 152-153.

- L135: Why only February and September? Is it a user option to select for other (or all months)? If so, please be clear what the difference is between options for the tool and what is being shown here as demonstration.

These sentences have been revised in lines 178-181. Our codes in the repository only show mean winter (February for the Arctic and September for the Antarctic) ice thickness and snow depth due to the limited observational months for the evaluation before 2007. The mean ice thickness and snow depth

differences of other months in both hemispheres can be provided for diagnosis in the future during other study periods when observational references are available.

- L187: Please describe the options for the user to select years for comparison

We have added a sentence in lines 231-232 to describe it.

- L204-205: Would the authors recommend freeboard be included in future CMIP model outputs for observational comparison? If so, include this in the conclusions.

The sea ice freeboard (mean height of the sea ice / snow interface above sea level) is already a requested variable of CMIP6 Sea Ice Model Intercomparison Project (see Notz et al. 2016, p.16, section C.2.5). At present, five CMIP6 OMIP models provide sea ice freeboard. We can expect more CMIP OMIP model outputs of sea ice freeboard in the future.

Notz, D., Jahn, A., Holland, M., Hunke, E., Massonnet, F., Stroeve, J., Tremblay, B., and Vancoppenolle, M.: The CMIP6 Sea-Ice Model Intercomparison Project (SIMIP): understanding sea ice through climate-model simulations, Geosci. Model Dev., 9, 3427–3446, https://doi.org/10.5194/gmd-9-3427-2016, 2016.

- L215/Table 2: separate dataset name and reference into separate columns

Done.

- L220: Perhaps I have misunderstood something about the calculations, but how can you determine the typical error to get the metric shown in Fig. 7b for Envisat without the second observational product (SnowModel-LG)

Because the observational data are not complete to calculate differences between two observational references, the typical errors of ice thickness and snow depth are calculated from the thickness and snow depth uncertainties of specific months from Envisat data. This information is given in lines 173-175.

- L272: Is the primary difference between OMIP1 and OMIP2 protocols the atmospheric model JRA55 vs. CORE-II? If so, I suggest it may be more clear to use "J" and "C" rather than 1 and 2.

Yes, we have changed the '/1' and '/2' into "/C" and "/J" in all figures.

- L272/Fig. 2: Are there any significant differences in patterns between the two observational products (NSIDC and OSI)? If not, why not just use the normal error relative to the mean between the two

products? I do not believe that the difference in comparison between the two products is a key point here, so I'm not sure it is useful.

The values are different compared to two observations and differences in patterns exist on the interannual variability of the Antarctic ice concentration compared to the OSI-450 data as shown in the fifth column of Fig. 2b. By providing the comparison to the two observations, we find that the overall ice concentration simulations (mean state, interannual variability, and trend) in both hemispheres are improved under OMIP2 protocol, which is not sensitive to the chosen observational reference and then robust. This is clarified in lines 300-304.

- L272/Fig 2: [suggestion] It may be helpful to show this after Figures 3 and 4 (showing specific metrics), so that these values have some context and introduction already

Figure 2 shows the ice concentration metrics. The specific metrics for seasonal cycle and anomalies of ice extent in Figs. 3 and 4 are shown in Fig. 5.

- L272/Fig. 2: Does "Ano" in figure refer to anomaly? (i.e. interannual variability?) This somehow needs to be made more clear, such as in the figure caption.

Yes, the 'Ano' refers to monthly anomalies. The six columns in Fig. 2 correspond to model performance metrics on the mean state, standard deviation (Std Ano) and trend (Trend Ano) of monthly anomalies of the Arctic and Antarctic ice concentration during 1980-2007. We have added the meaning in the figure caption.

- L272/Fig. 2: [suggestion] add some vertical space between NH and SH. "NH" and "SH" may be sufficient (rather than "North" and "South"), and would save some room

Thanks for your suggestions. A dashed grey line is added between NH and SH to distinguish two hemispheres. We have used 'NH' and 'SH' in the modified figures.

- L290/Fig. 3: The multi-model mean line is hard to distinguish. Consider using a thicker or brighter/more distinctly colored line.

Thanks for pointing this out. We have moved the multi-model mean lines and observational marks into another figure to help distinguishing the differences.

- L310/Fig. 4: If possible, it would be helpful to explicitly label "std" and "trend" on these plots to demonstrate where values in Fig. 2 are coming from.

The standard deviation and trend have been labeled in the revised Fig. 4. The plots in Fig. 4 are monthly anomalies of ice extent from the observational mean, OMIP1 and OMIP2 model mean, which are not used for the calculation in Fig. 2. The ice concentration metrics in Fig. 2 are derived from the monthly anomalies of ice concentration and the methods in calculating the standard deviation and trend are close.

- L316: Interesting. What are the implications of this? Should typical error not be used in this case, or used with caution? Are there similarities in how some products are derived that result in this metric having less utility?

The typical error of the interannual ice extent variability from two observations is very low. This reveals that the Antarctic ice extent variability is very close from two observations. The large values on the fifth columns of ice extent metrics reveal that large bias exists in the simulation of the Antarctic interannual ice extent variability in OMIP models compared to the observations. The typical error is treated as a benchmark and needs to be used to evaluate these ice variables. We can evaluate model skills based on this benchmark.

- L341-2/Fig. 5: Separate by product in panel (c) to be consistent with other Figure. (Unless products are combined with averaging, as suggested in comment above)

Thanks for your suggestion. Figure 5c shows the mean state ice edge location metrics compared to NSIDC-0051 and OSI-450 data. In order to save space, these two comparisons are put together. By providing the comparison to the two observations, we can prove that the results are not sensitive to the chosen observation and then robust.

- L336/Fig. 5: Perhaps include in subplot title a summary of what is being evaluated, such as "Extent: models vs. NSIDC"

We have included this in the subplot titles.

- L366: I'm not sure I understand how you can have IIEE for the observational product OSI-450. Is this rather the observational products? In that case, should it be called the "typical error" here?

Yes, it's the Integrated Ice-Edge Error (IIEE) between the OSI-450 and NSIDC-0051 data. We have modified the labels in Fig. 6 to make it clear.

- 8/Fig. 9: Are these annual means/using all months? Please specificy period evaluated in figure captions.

Thanks for your suggestion. The vector correlation coefficients are derived from monthly mean ice vector data during 2003-2007. We have added this information in the figure captions.

- L489-499: Are the plots the ones that are used in the manuscript and/or the appendix? Please clarify

We have revised this sentence in lines 560-561. SITool (v1.0) produces spatial maps (Figs. 8, 9 and Figs. A1-A12 in Appendix A) and time series diagrams (Figs. 3, 4, 6) that can be consulted by the expert to understand the origin of one particular metric value.

- It would be helpful to provide example of completed code run and diagnostic plots in the repository on Github

Thanks for pointing this out. We have provided the completed code run and the corresponding figures in the repository on Github.

**Reviewer 3:**

Review of "SITool (v1.0) - a new evaluation tool for large-scale sea ice simulations: application to CMIP6 OMIP" by Xia Lin, François Massonnet, Thierry Fichefet, Martin Vancoppenolle (gmd-2021-99).

[General comments]

This paper introduces an evaluation tool for sea ice simulation and presents its application to CMIP6-OMIP simulations available through ESGF. I think that such a tool will become a valuable asset for the climate/sea ice modeling community and such activities should be strongly encouraged. Calculation methods of metrics are well described and the evaluation using this tool is well presented. The comparison between OMIP-1 and OMIP-2 simulations, which use different surface atmospheric forcing dataset, is timely and should be highly appreciated. However, I think that some discussion would be needed for the proposed method for the evaluation of interannual variability and trend as commented below.

[Specific comments]

Metrics are proposed for the monthly mean state, interannual variability, and trend, with each metric basically using common calculation method: difference between simulation and observational reference is scaled by observational uncertainty based on the difference between two observational datasets. For me, applying this method to the monthly mean state was understandable, but it was somewhat difficult to interpret the specific values of metrics for interannual variability (standard deviation of monthly anomalies) and trend. If I was to evaluate interannual variability of a simulation, I would like to know the size of the standard deviation of monthly anomalies relative to that of observational reference. Specifically, I think that

the metrics would be easier to interpret if the standard deviation was scaled by the that of an observational reference and the range of values obtained by applying different observational references were presented. The same argument would be applied to trends and in this case the signs of trends could be also evaluated. I would like to ask the authors to explain the background behind the choice of the current method.

I would like to add that it would be useful and clear if the calculation methods are presented using mathematical formulas.

Thanks for your comments. The method you suggested to evaluate interannual variability of a simulation makes sense and is defendable but (1) it breaks the consistency with how the metric is calculated for the mean state and (2) it is not clear in your proposition where we can introduce observational uncertainty and determine the typical error, which is a need we strongly agreed upon when designing our tool. In our manuscript, the standard deviations of monthly anomalies from model outputs and the two observations are computed separately, and then the differences between model outputs and the observation are scaled by the typical error to evaluate the model skill. The typical error is the differences between two observations and treated as a benchmark. We can evaluate model skills based on this benchmark. The same idea applies for the trend evaluation. We have added these contents in lines 103-104 and in section 2.1.1.

The mathematical formulas have been added to clarify the calculation in section 2.1.1.

[Technical corrections]

L135, 150, 164: Why equal weight is used for these metrics?

The ice concentration is the percentage of sea ice in a grid cell, which is defined by the grid cell area. For the ice thickness, snow depth and ice drift metrics, they are not defined by the grid cell area. Then equal weight is used for those metrics.

L184: "the influence model resolution" should read "the influence of model resolution".

We have modified this text.

L286: "exits" should read "exists".

We have modified this text.

L288: "without reduction"… I could not understand the meaning of this phrase in the sentence.

We have modified this text in lines 331-332.

Figure 3: It was difficult for me to distinguish the lines. I would suggest the figures to be separated for OMIP-1, OMIP2, and their means, that is, into the total of six figures.

Thanks for your suggestion. We have separated the figures and moved the multi-model mean lines and observational marks into another figures to help distinguish the differences.

Yours sincerely,

Xia Lin, François Massonnet, Thierry Fichefet, Martin Vancoppenolle